# The Truth is in There:
# Improving Reasoning in Language Models with Layer-Selective Rank Reduction

**Pratyusha Sharma**[1]     **Jordan T. Ash**[2*]  **Dipendra Misra**[2*]
[1]Massachusetts Institute of Technology   [2]Microsoft Research NYC

## Abstract

Transformer-based Large Language Models (LLMs) have become a fixture in modern machine learning. Correspondingly, significant resources are allocated towards research that aims to further advance this technology, typically resulting in models of increasing size that are trained on increasing amounts of data. This work, however, demonstrates the surprising result that it is often possible to significantly improve the performance of LLMs by selectively removing higher-order components[1] of their weight matrices. This simple intervention, which we call LAyer-SElective Rank reduction (LASER), can be done on a model after training has completed, and requires minimal additional parameters and data. We show extensive experiments demonstrating the generality of this finding across language models and datasets, and provide in-depth analyses offering insights into both when LASER is effective and the mechanism by which it operates[2].

## 1 Introduction

Since their original release, Transformer-based LLMs have been shown to be remarkably proficient on a wide array of important machine learning tasks. Their underlying Transformer architecture has become state-of-the-art for modeling and reasoning about natural language, and has shown promise in domains such as computer vision (Dosovitskiy et al., 2021) and reinforcement learning (Chen et al., 2021) as well.

Contemporary instantiations of Transformer architectures are infamously large, typically requiring tremendous compute resources for both training and inference. This is by design, as Transformers trained with more parameters and data are demonstrably more capable than their slimmer predecessors—often by a significant margin (Brown et al., 2020; Touvron et al., 2023). Still, a growing body of work suggests that Transformer-based models, and neural networks more generally, do not require all fitted parameters to retain their learned hypotheses. While it seems helpful to be massively over-parameterized at train time (Hinton et al., 2014; Bengio et al., 2005), it is well-known that these models can be drastically pruned at test time; neural networks can often have well over 90% of their weights removed without any significant degradation in performance (Frankle and Carbin, 2019). The discovery of this phenomenon bolstered interest around the relationship between generalization and over-parametrization (Zhang et al., 2017), and spawned research in developing pruning strategies that lend themselves to efficient model inference (Molchanov et al., 2017).

This paper presents a surprising finding, that careful pruning done at specific layers of Transformer models can produce significant boosts in performance on some tasks. We describe LAyer SElective Rank reduction (LASER), an intervention that removes higher-order components of learned weight matrices as identified by singular value decomposition. This reduction is performed in specific weight matrices and layers of the Transformer model. In line with previous work, we find that many such matrices can be significantly reduced, and that performance degradation is often not observed until well over 90% of components are entirely removed. However, unlike what is found in previous work, we find that these reductions can produce drastic improvements in accuracy, as measured by

---

[*]Asterisk indicates equal advising, $\alpha$-$\beta$ listed. Correspondence to `pratyusha@mit.edu`.
[1]Higher-order components are singular vectors with smaller singular values.
[2]Code and website: https://pratyushasharma.github.io/laser/

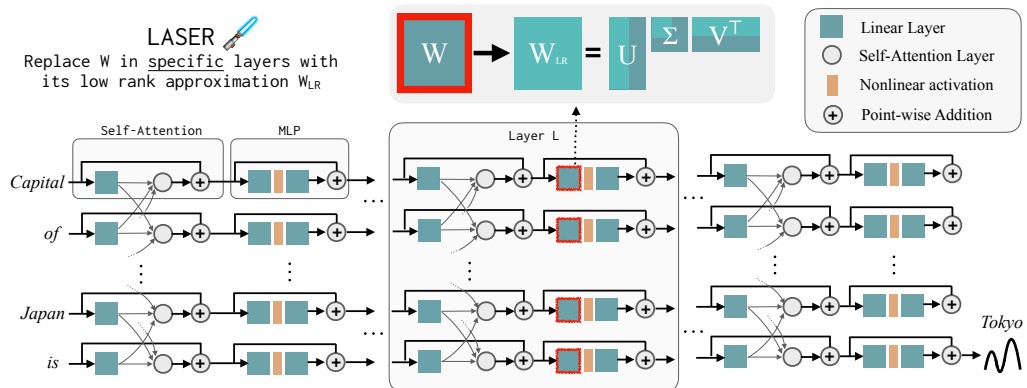

Figure 1: LAyer SElective Rank reduction (LASER) replaces a specific weight matrix $W$ of the Transformer model with its rank-$k$ approximation, $W_{LR}$, and observes the change in the model's behavior. We find that this low-rank approximation, especially for MLP weights in the latter layers of the model, often offers surprising benefits to model performance.

various well-studied reasoning benchmarks in NLP. Even better, our discovery appears to not be limited to natural language, with performance gains also found in reinforcement learning.

This paper analyzes the relationship between the model's training data and samples that benefit from LASER. We find that improvements in the model's performance predominantly correspond to information less frequently present in the model's training data, suggesting that LASER offers a kind of denoising procedure that makes weakly learned facts accessible. We separately observe that LASER affords increased robustness to paraphrases on previously correct questions.

Additionally, we attempt to reason about what is being stored in high-order components, such that their removal boosts performance. For questions correctly answered only after LASER, the original model (without any intervention applied) predominantly responds with high-frequency words such as "the," "of," etc—generations that are not even the same semantic type as the correct answer. However, after some amount of rank reduction, the model's answer flips to be correct. To understand this, we look at what the components removed by LASERencode on their own; we approximate the weight matrix using only its higher-order singular vectors. We find that these components describe either a different response of the same semantic category as the correct answer or generic high-frequency words. Seemingly, when noisy, higher-order components are combined with low-order components, their conflicting responses produce a sort of "average answer," which is likely incorrect.

Figure 1 visualizes the Transformer architecture and the procedure followed by LASER. Here, the weight matrix of a Multi-Layer Perceptron (MLP) layer is replaced with its low-rank approximation.

## 2 RELATED WORK

To our knowledge, this paper is the first to identify that carefully selected rank reduction interventions can boost Transformer performance. Still, there is a wide array of works that study related questions, including how facts are stored in LLMs and how to best compress neural networks.

**How facts are stored.** Studies probing model representation for the presence of select properties of entities (Ettinger et al., 2016; Adi et al., 2016; Hupkes et al., 2018; Conneau et al., 2018) argue that models store factual information across different layers fashion. However, there is conflicting evidence on how this information is organized and utilized in constructing answers in large language models. Some theories posit that information about different entities is locally stored as two-layer, key-value memory in MLP sections of Transformer models (Geva et al., 2021), which are thereafter copied over through latter layers by the self-attention modules (Elhage, 2021). Meng et al. (2022) propose a procedure to trace and edit local, entity-specific information to map to distinct "impossible" outputs, supporting the locality theory. These theories are further supported by the phenomenon of "early exiting," where the representation at an intermediate layer can be directly used with the terminal head of the model to correctly generate an output (Zhao et al., 2021). In contrast, Hase et al.

(2023) have observed that information about overlapping entities or entity relations can be modified by editing a variety of layers in the model architecture, and therefore, that facts are stored across layers in a fragmented fashion. This paper makes no specific claims regarding locality, but rather demonstrates that higher-order components of a weight matrix introduce noise in decision making, and that considering only lower-order components may make correct answers accessible.

**Model compression.** Neural network pruning methods have found that models could be significantly pruned (often removing over 90% of parameters) with very little drop in accuracy, significantly reducing the storage requirements of the model (LeCun et al., 1989; Hassibi and Stork, 1992; Han et al., 2015; Li et al., 2017; Frankle and Carbin, 2019). There have also been approaches that prune these models in a structured manner, to facilitate improvements in inference time (Molchanov et al., 2017). The existence of sparse sub-networks (Frankle and Carbin, 2019; Hoefler et al., 2021) has been found to be true for convolutional, fully connected, and Transformer models (Lv et al., 2023; Murty et al., 2023). While Jin et al. (2022) find that model generalization can be improved by pruning and then refitting parameters, generalization improvements are only observed upon model retraining. To our knowledge, model pruning techniques have always done an indiscriminate reduction across all parameters, without targeting any specific layers—leading to predictive performance either staying fixed or decreasing (Frankle and Carbin, 2019). In this work, however, we find that the effect of reduction in accuracy is non-uniform across different layer types, and a model's generalization can be improved by *selective* pruning alone; no additional training is necessary. Roughly, we find that performance degradation can be produced by rank-reducing early layers, while significant performance benefits are typically available by pruning later layers.

**Low-rank approximations of weight matrices.** Most pruning methods reduce parameters in order of their absolute magnitude (Frankle and Carbin, 2019). An alternative approach, however, is to reduce the rank of its constituent weight matrices, keeping the top $k$ components found by SVD. While matrices of neural models, including Transformer models, have been found to be well-approximated using this approach, where markedly reduced versions of a model can preserve much of its original behavior, research has shown that performance eventually declines as the severity of the intervention increases (Lv et al., 2023; Hajimolahoseini et al., 2021; Yu et al., 2017). Note that these reductions are typically done universally, removing the same number of components in every weight matrix in the model. In contrast to these findings, we show that a targeted rank reduction, even affecting just a single weight matrix, can offer benefits to the accuracy of Transformers.

**Model distillation and low-rank training.** Ba and Caruana (2014) and Hinton et al. (2014) have trained smaller networks to mimic the behavior of larger networks, suggesting neural networks might be significantly over-parametrized and can be replaced with leaner alternatives. To our knowledge, no report of an improvement in the model's predictions as a consequence of this procedure has been shown. Yang et al. (2020) have enforced low-rank-ness of weight matrices for the purposes of memory efficiency, but the resulting models fail to achieve performance equivalent to their over-parametrized counterparts. The result suggests that over-parametrization is helpful for the identification of well-generalizing parameters by SGD (Bengio et al., 2005; Hinton et al., 2014).

## 3 PRELIMINARIES

Here we review basic notation and describe the core components of our study.

**Transformer Architecture.** We provide a concise description of a typical Transformer architecture that is relevant to our analysis. A Transformer is composed of $L$ layers of Transformer blocks, each adjusting their input representation from the previous block using two sequential steps: a self-attention mechanism to consider information across time steps, and a feed-forward network to process information within each time step. [3] Importantly, a Transformer architecture has the following weight matrices $\mathcal{W} = \{W_q, W_k, W_v, W_o, U_{in}, U_{out}\}$ for each layer, in addition to an embedding matrix for embedding input tokens, a projection weight matrix applied after the final layer before performing a softmax, and some additional weights associated with layer normalization. This work primarily considers matrices in $\mathcal{W}$, and intervenes by replacing them with low-rank approximations.

---

[3] Various Transformer models often have small differences in how these transformations are implemented. Our goal is not to provide a full survey of these details but to capture essential terminology for our results.

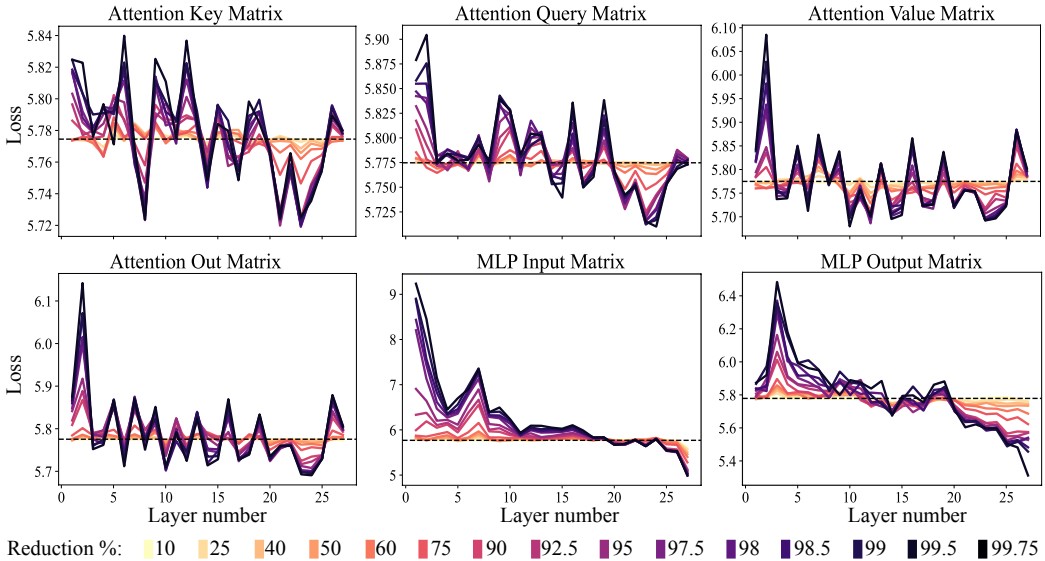

Figure 2: The effect of rank reduction across different layer types is not uniform. Here we show the effect of rank reduction for GPT-J as studied on the CounterFact dataset. The dashed line is the um-modified network's loss. In the attention layers (key, query, value, out matrices), while it is clear matrices could be significantly rank-reduced without damaging the learned hypothesis, there is very little performance increase. However, for the multi-layer perceptron (MLP) layers, rank reduction goes from uniformly harming to improving the model's performance (around layer 20).

**Rank-$r$ Approximation and SVD.** Given a matrix $W \in \mathbb{R}^{m \times n}$ and $r \in \mathbb{N}$, a rank-$r$ approximation problem requires finding a matrix $\widehat{W}$ that minimizes $\|W - \widehat{W}\|_2$ and satisfies $\texttt{rank}\left(\widehat{W}\right) \leq r$. Eckart–Young–Mirsky theorem provides an optimal solution of this problem using Singular Value Decomposition (SVD) (Eckart and Young, 1936). Formally, an SVD of a matrix $W$ is given by $W = U\Sigma V^\top$ where $U = [u_1, u_2, \cdots, u_m] \in \mathbb{R}^{m \times m}$ and $V = [v_1, v_2, \cdots, v_n] \in \mathbb{R}^{n \times n}$ and $\Sigma \in \mathbb{R}^{m \times n}$. The column vectors of $U$ and $V$ constitute an orthonormal basis of $\mathbb{R}^m$ and $\mathbb{R}^n$ respectively, and $\Sigma$ is a diagonal matrix whose entries are the singular values of $W$ in descending order. One can also express the SVD of $W$ as $W = \sum_{i=1}^{\min\{m,n\}} \sigma_i^\downarrow(W) u_i v_i^\top$. According to Eckart–Young–Mirsky theorem, the matrix $\widehat{W} = \sum_{i=1}^{r} \sigma_i^\downarrow(W) u_i v_i^\top$ is an optimal solution to the rank-$r$ approximation problem for any given desired rank $r \leq \min\{m, n\}$.

In this work, we will use the word **higher-order components** to refer to entries in the SVD corresponding to the components with smaller singular values. These components are removed by `LASER`. The term **lower-order components** is used to refer to singular vectors corresponding to large singular values. These components are kept in a low-rank approximation of the matrix.

## 4   LAYER SELECTIVE RANK REDUCTION (LASER)

In this section, we formally describe the `LASER` method. A single-step `LASER` intervention is defined by three quantities, $(\tau, \ell, \rho)$, consisting of a parameter type $\tau$, layer number $\ell$, and rank reduction $\rho$ respectively. These values together describe which matrix will be replaced by its low-rank approximation and how severe the approximation will be. The parameter type categorizes the matrix type in which we will intervene. We focus on the matrices in $\mathcal{W} = \{W_q, W_k, W_v, W_o, U_{in}, U_{out}\}$ which consist of the matrices in the MLP and attention layers. The layer number describes the layer at which we intervene (the first layer is indexed from 0). E.g., the Llama-2 has 32 layers and so $\ell \in \{0, 1, 2, \cdots 31\}$. Finally, $\rho \in [0, 1)$ describes what fraction of the maximum rank should be preserved upon doing its low-rank approximation. For example, let $\tau = U_{in} \in \mathbb{R}^{d \times d}$, then the maximum rank of this matrix is $d$. We replace it with a rank $\lfloor \rho \cdot d \rfloor$-approximation.

Figure 1 shows an example of LASER. In this figure, we have $\tau = U_{in}$ and $\ell = L$ indicating that we update the weight matrix in the first layer of MLP in the Transformer block of the $L^{th}$ layer. The other parameter (not shown in the figure) controls the $k$ in the rank-$k$ approximation.

LASER throttles the flow of certain information in the network, which surprisingly can produce significant performance benefits. These interventions can also be easily composed—we can apply a set of interventions $\{(\tau_i, \ell_i, \rho_i)\}_{i=1}^m$ in any order. The LASER approach is to simply search over interventions of this type, and to exercise the modification that offers the greatest benefit. There are many other ways in which one can combine these interventions, however, and we defer this to future work.

## 5 EXPERIMENTS

This section studies the consequences of LASER throughout various layers of the Transformer architecture. We first perform a motivating analysis of the CounterFact (Meng et al., 2022) question-answering dataset in conjunction with a pretrained GPT-J model (Wang and Komatsuzaki, 2021), and investigate the performance of the model and its variability as we search over potential interventions. We then examine the effect of LASER across different models, datasets and modalities.

**GPT-J, CounterFact and PILE.** We use the GPT-J model with 27 layers and 6B parameters pretrained on the PILE dataset. The first part of our analysis focuses on GPT-J, largely because its training data is publicly available. We evaluate the model's behavior on the CounterFact dataset, which consists of samples organized as (subject, relation, answer) tuples and three paraphrased prompts for each question. For example, (Danielle Darrieux, mother tongue, French).

### 5.1 A THOROUGH ANALYSIS WITH GPT-J ON THE COUNTERFACT DATASET

Figure 2 shows the result of applying various amounts of rank reduction to each matrix in the Transformer architecture on the classification loss for this dataset. These plots are grouped, such that each sub-figure corresponds only to the indicated type of weight matrices. Note that each Transformer layer consists of a small, two-layer MLP. The constituent input and output matrices are shown separately. Different colors indicate different percentages of removed components.

The attention plots in this figure exemplify what is already known about these models: weight matrices can be drastically reduced without much degradation in model performance. The more interesting result, however, is in the MLP layers. Here, not only can matrices be rank-reduced without degrading classification performance, but large performance improvements are possible by reducing later layers of the model. This trend is most stark in the input matrix of the MLP. While there are gains with LASER in the attention layers too, the benefits are typically smaller. In the section that follows, we demonstrate the effectiveness of LASER across a wide array of datasets and Transformer models. Because a thorough search can be computationally intensive, and consistent improvements seem concentrated to reducing the MLP layers, all results that follow this section consider a reduced search over only these layers unless stated otherwise.

**Improved accuracy and robustness to paraphrases.** The CounterFact dataset is used to test the model's factual knowledge of data from Wikipedia. Since GPT-J is trained on PILE, whose contents include Wikidata, different facts in CounterFact are part of the model's training data, albeit in different quantities. As all answers are a single token in this setting, we compute top-k accuracy based on whether the correct answer is in the top-k predicted tokens. As seen in Figure 2 and Table 1, we find that the model's top-1 accuracy on facts in CounterFact increases from 13.3% to 24.1% when reductions are done on a single layer. It is important to note that these improvements are a result of rank-reduction alone, and do not involve any further training or fine-tuning of the pre-trained GPT-J model. Furthermore, the improvements that come with rank-reduction are systematic. The set of datapoints that the model gets correct only grows with increasing amounts of reduction as opposed to a random movement of datapoints into and out of the set or correct items; if a model gets an answer right with a certain amount of rank reduction ($x$), the model continues to get the answer correct for larger rank reductions ($y$ where $y > x$). We evaluate the model's robustness to paraphrases by computing the percentage of datapoints where the model gets all paraphrases of a given question correct. For datapoints that the model already gets correct, the model's robustness to paraphrases also improves with LASER by roughly 24.8 percentage points.

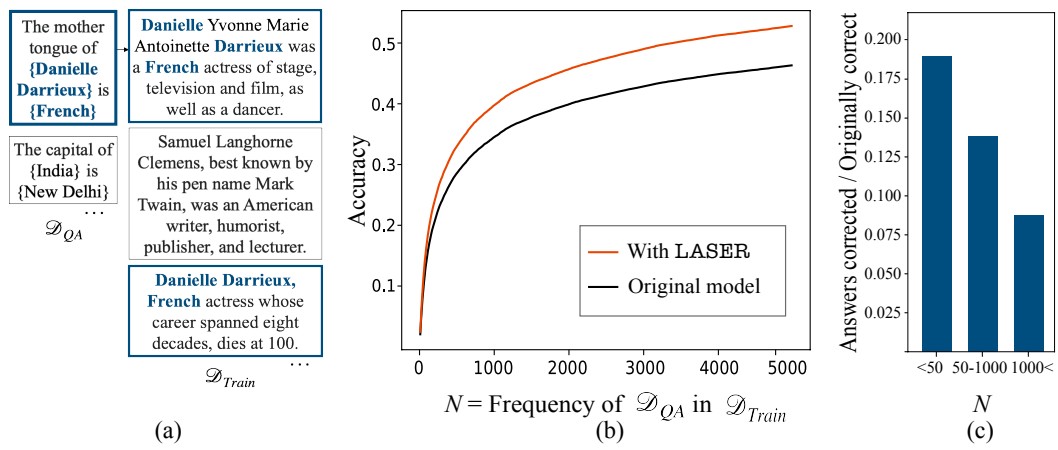

(a)

(b)

(c)

Figure 3: Which datapoints benefit from LASER? We analyze how frequently in the training data "corrected" facts occur. GPT-J is an ideal test bed for such analysis since its training data ($\mathcal{D}_{Train}$), the PILE dataset, is publicly available. (a) For GPT-J evaluated on CounterFact ($\mathcal{D}_{QA}$) we retrieve all the datapoints in $\mathcal{D}_{Train}$ that contain a mention of both the entity of interest and the answer that correspond to each sample in $\mathcal{D}_{QA}$. (b) A plot depicting the cumulative top-10 accuracy of the model on all datapoints that occur in the training data less than or equal to the frequency indicated on the x-axis. Here we show accuracy with and without LASER. (c) The largest boost in performance occurs for low-frequency samples. This bar chart displays the amount of boost offered by LASER for data binned by the frequency with which corresponding facts occur in $\mathcal{D}_{Train}$. Maximal improvements in accuracy are from datapoints that have less-frequent occurrences in training data.

**Effect on language modeling and fluency.** While the model's factuality improves, does the reduction affect the model's performance on other metrics? To understand this, we evaluate the model's perplexity, i.e., its original training objective, on its training data. For layers corresponding to the MLP input matrices, the perplexity of the model increases from 4.8 to 5.0, showing that the language modeling objective is indeed slightly effected. For the MLP output layers, the perplexity of GPT-J on PILE increases from 4.8 to 4.9 with LASER. It may be possible to fix this small degradation by calibrating the temperature of the model.

**Composing reductions across layers.** We find that even further improvements in the model's performance can be made by performing different amounts of rank reduction across several layers. This is done by greedily searching over $(\tau, \ell, \rho)$ starting from the largest $\ell$ and smallest $\rho$. To speed things up, here we do this search only over MLP layers, as this is where the largest improvements are typically found. Consistent with other experiments, the search is done on a validation set, and results are reported on the test set. On CounterFact, the 0-1 accuracy of the base GPT-J model is 13.1%. After doing the best single-step LASER the model's accuracy improved to 24.0%. Performing LASER across different layers improved the top-10 accuracy to 29.2%, a 5.2% absolute improvement in accuracy over performing LASER on a single layer. The results of the combinatorial search across different $\ell$ and $\rho$ values can be seen in Figure 4.

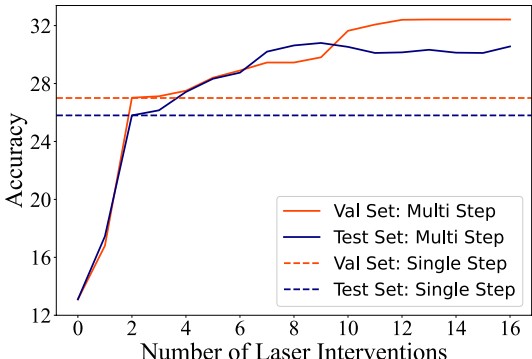

Figure 4: Composing LASER operations across multiple layers further enhances model performance. Here we show how accuracy improves using a simple compositional strategy for both validation data, which was used to identify each $(\tau, \ell, \rho)$, and held-out test data.

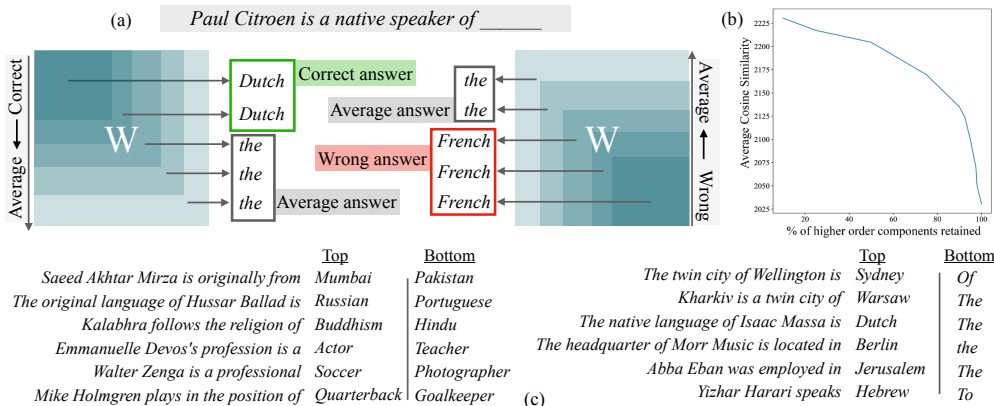

Figure 5: (a) [*Left*] LASER approximates learned matrices by its lower-order components. We find that for samples where the model's predictions improve after LASER, if we instead use the entire matrix (including higher-order components), the model often predicts only "generic" words. (a) [*Right*] To understand what these higher-order components encode, we approximate the learned weight matrix with the higher-order components instead. We find that these higher-order components sometimes encode the correct semantic type of the answer but the incorrect response. (b) Analytically, computing the semantic similarity (cosine distance between the true answer and answers generated by the bottom k% of the singular vectors) shows that, on average, the answer computed by higher-order components is more similar to the real answer. (c) Shows examples from the dataset and corresponding answers computed by the top fraction and bottom fraction of components.

### 5.1.1 WHICH FACTS IN THE DATASET ARE RECOVERED BY RANK REDUCTION?

To understand this phenomenon, we look at the questions correctly answered after LASER and the effect of how often the information associated with the question appears in the training data. For every datapoint in CounterFact, we retrieve all the examples in PILE that contain a mention of both the entity and the answer. We then compute how often information associated with each evaluation question appears in the training data. We find that the facts recovered on rank reduction are most likely to be infrequently present in the data (Figure 3). Here, "Originally correct" describes samples that are correctly classified even without any intervention. "Answer-corrected" refers to questions the model gets correct only after intervening with LASER.

### 5.1.2 WHAT ARE HIGHER-ORDER COMPONENTS STORING?

We saw above how retaining the lower-order components improves model performance on the task of open-ended question answering. We find that for the task of question answering, improvements often come on questions whose answers are supported by less frequently occurring data in the training set. While it is clear that eliminating the higher-order components "denoises" the model and helps recover "hidden," less-frequent information, it is not clear what the higher-order components are representing such that that their removal improves performance. This section studies this question using the CounterFact dataset and GPT-J.

To understand what higher-order components are representing, we approximate the final weight matrix using its higher-order components (as opposed to approximating it using its lower-order components as done by LASER) as shown in Figure 5(a). Following this, we analyze how the model's behavior changes on data points that GPT-J originally gets incorrect but are flipped to being correct upon performing LASER.

First, we note that when the original, unmodified model does not answer these questions correctly, it often responds with common words, such as "a," "the," "of," and other highly frequent tokens. After performing LASER, where we retain only the top-k components, the model's answers to these questions flip from generic words to the correct entity. For the same datapoints, when we approximate the model by instead retaining the higher-order components, we find that the model either predicts incorrect entities that are of the same semantic type as the correct answer or high-frequency tokens

| Dataset | | Model Name | | | | | |
|---|---|---|---|---|---|---|---|
| | | Roberta | | GPT-J | | LLama2 | |
| | | | LASER | | LASER | | LASER |
| CounterFact | Acc | 17.3 | **19.3** | 13.1 | **24.0** | 35.6 | **37.6** |
| | Loss | 5.78 | **5.43** | 5.78 | **5.05** | 3.61 | **3.49** |
| HotPotQA | Acc | 6.1 | **6.7** | **19.6** | 19.5 | 16.5 | **17.2** |
| | Loss | 10.99 | **10.53** | 3.40 | **3.39** | 3.15 | **2.97** |
| FEVER | Acc | 50.0 | **52.3** | 50.2 | **56.2** | 59.3 | **64.5** |
| | Loss | 2.5 | **1.76** | **1.24** | 1.27 | 1.02 | **0.91** |
| Bios Gender | Acc | 87.5 | **93.7** | 70.9 | **97.5** | 75.5 | **88.4** |
| | Loss | **0.87** | 1.13 | **3.86** | 4.20 | 3.48 | **2.93** |
| Bios Profession | Acc | 64.5 | **72.5** | 75.6 | **82.1** | 85.0 | **86.7** |
| | Loss | **4.91** | 6.44 | **4.64** | 4.91 | 4.19 | **4.05** |
| TruthfulQA | Acc | 56.2 | 56.2 | 54.9 | **55.6** | 50.5 | **56.2** |
| | Loss | 1.60 | **1.42** | 1.02 | **1.01** | **0.95** | 1.04 |
| BigBench-Epistemic Reasoning | Acc | 37.1 | **41.8** | 37.1 | **38.3** | 44.8 | **63.4** |
| | Loss | 9.39 | **6.80** | 0.74 | **0.62** | 0.78 | **0.73** |
| BigBench-WikidataQA | Acc | 28.0 | **30.7** | 51.8 | **65.9** | 59.5 | **62.0** |
| | Loss | 9.07 | **7.69** | 3.52 | **2.86** | 2.40 | **2.31** |

Table 1: The effect of LASER intervention on eight natural language understanding datasets. We find the best LASER intervention for each model and task using accuracy/0-1 on a validation set and report its performance on a held-out test set. We notice that in some cases, while the model's accuracy improves, its loss slightly degrades.

such as "a," "the," and "of," as shown in Figure 5(c). However, as we systematically include the lower-order components, the model's output changes to predicting frequent tokens. To investigate this systematic degradation, we measure the average cosine similarity of the "true" answer with respect to the predicted answer when the matrix is approximated with different amounts of higher-order components, as shown in Figure 5(b). The average cosine similarity between the predicted answer worsens, demonstrating this effect.

We hypothesize that these matrices often encode multiple conflicting responses, and that when all components are used they clash to produce a generic token. Removing the higher-order components, which anecdotally appear to often capture incorrect responses of the correct type, resolves this internal conflict and allows the model to respond accurately.

## 5.2 HOW GENERALLY DOES THIS HOLD?

We study the generality of our findings by applying three different LLMs to a variety of natural language understanding tasks.

**Natural Language Understanding Tasks.** We evaluate model performance before and after LASER on seven datasets, including CounterFact (Meng et al., 2022), HotPotQA (Yang et al., 2018), FEVER (Thorne et al., 2018), Bias in Bios (De-Arteaga et al., 2019) [Gender and Profession], TruthfulQA (Lin et al., 2021), BigBench-Epistemic Reasoning (Bowman et al., 2015) and BigBench-WikidataQA. These datasets evaluate different aspects of language understanding problems. CounterFact, Fever, and Bigbench-Wiki data test a model's world knowledge and factuality. Bias in Bios benchmarks model bias by predicting the gender and profession of a person given a short biography. We define Bios Gender as the gender prediction problem in Bias in Bios, and Bios Profession as the profession prediction problem. HotPotQA provides a more challenging open-ended question answering task with long answers containing many tokens. The Epistemic Reasoning dataset from Big Bench Hard (BBH) tests a model's logic and reading comprehension. Finally, TruthfulQA tests an LLM's truthfulness.We use 20% of the dataset as validation set and select the best LASER hyperparameters $(\tau, \ell, \rho)$ using this validation set. We report results on the remaining 80% of the dataset with the chosen hyperparameter. The models used for the task of question answering include, Roberta (Liu et al., 2020), GPT-J (6B) (Wang and Komatsuzaki, 2021), and LLAMA2 (7B) (Touvron et al., 2023). Details regarding datasets and how they were used can be found in Appendix A.

**Evaluation metrics.** For each of these tasks, we evaluate the model's performance using

1. **Generation accuracy.** We generate a sequence of $N$ tokens using the LLM and then report one if the answer text is in the generated text and zero otherwise,

2. **Classification accuracy**. If the answer lies in a small set of potential values, like in a standard classification problem, we consider a response correct if it puts more probability mass on the correct answer than on any of the other candidates.

3. **Loss.** We report the log-loss on held-out data. For datasets with a small set of possible labels we report the (**acc**) classification accuracy, and for others we use generation accuracy.

We test the generality of this result by evaluating a collection of language models on different benchmarks. As seen in Table 1, we find that even severe reductions result in no deterioration in the model's accuracy and can lead to improvements in their performance. The amount of reduction required differs from model to model.

## 5.3 NON-TEXT DOMAINS

To understand if this phenomenon is effective outside of question answering in the textual domain, we evaluate the effect of rank reduction on a reinforcement learning agent.

**Policy learning.** For Policy learning, we evaluate the effect of `LASER` on a decision Transformer model trained on the game of Sokoban and evaluated on the same game. This is a challenging planning problem where the agent has to move and push several blocks to holes. The task is completed when all blocks are on top of holes. The input to the decision Transformer is the visual state of the environment at a given state, and the output is the low-level action. We find that for a decision Transformer trained on Sokoban, models solved

| Model Name | Acc. | Return |
|---|---|---|
| Transformer | 50.67 | 0.575 |
| **with** LASER | **53** | **0.965** |

Table 2: Effect on `LASER` on a 6-layer Decision Transformer agent. The base model is trained and evaluated in a challenging $10 \times 10$ Sokoban domain.

3% more tasks with `LASER` (Table 2). Details of the experiment can be found in Appendix B.

Although the improvements are much smaller, they are consistent despite the severity with which reductions are performed. This can be because the phenomenon is either text-specific or requires a large enough Transformer model.

## 6 CONCLUSION AND DISCUSSION

This paper describes `LASER`, a phenomenon where performing a low-rank approximation of specific layer types at specific layers of the transformer block can improve the performance of LLMs on the task of question answering. We find this to be true across five different datasets and three different language model models. Furthermore, the resulting `LASER` reductions are extreme. The matrices are reduced at times to 99% of their original rank, which is much lower than their effective rank (C.1). However, despite extreme reductions, the performance of the model on tasks continues to improve. We also observe performance gains for a decision Transformer in an embodied domain. We find that the largest improvements in the model accuracy correspond to information that is less common in the training data and that `LASER` jointly makes the model more robust to paraphrases of the questions. We further found that the higher-order components of some of these matrices encode either high-frequency words or alternate answers of the same semantic type as the correct answer. These noisy, higher-order components can overpower the stable lower-order components and result in the model answering questions incorrectly. In these cases, performing `LASER` acts as a denoising technique and reduces the internal conflicts in potential responses.

Despite this analysis, the success of `LASER` requires further study. Learning (i) why higher-order components in weight matrices accumulate noisy answers in the course of training, (ii) the effect of model architecture and other structural choices on the occurence of this phenomenon and (iii) why this is specifically true for later layers in the MLP is important to not only for our understanding of the success of `LASER`, but for understanding the behavior of large language models more generally.

ACKNOWLEDGEMENTS

This work was done when PS was an intern at Microsoft Research, New York City, with DM and JTA. The authors would like to thank Minyoung Huh, Shikhar Murty, Han Guo, Cyril Zhang, David Bau, Jacob Andreas, Antonio Torralba, and John Langford for helpful discussions and feedback.

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

## A   DATASET DETAILS

**CounterFact.**   The CounterFact dataset is derived from the PARAREL dataset (Elazar et al., 2021) and contains knowledge tuples of the kind $t^c = (s, r, o^c)$, where $s$ is the subject, $r$ is the relation and $o$ is the object. These tuples are constructed using entities listed in Wikidata. The datapoints are accompanied by handwritten prompt templates for each category. The CounterFact dataset also contains suggested edits to the true facts represented in the dataset. For this study, the set of counterfactual edits are not used.

**PILE.**   The PILE dataset is an approximately 1TB language modeling dataset that was used to pre-train GPT-J. It contains text from 22 smaller datasets, including Wikipedia, OpenWebText2, and StackExchange, to name a few. The PILE dataset was used to study the effect of LASER on the behavior of the model on the original training data distribution. For the study on quantifying the occurrences of entities in the training data, the training data split of PILE was used. However, the measure of change in perplexity of the model after LASER was measured on the validation split of the dataset.

**HotpotQA.**   We use the HotPotQA dataset available on HuggingFace. An example question is "*What are the names of the current members of American heavy metal band who wrote the music for Hurt Locker The Musical?*" and the answer is "*Hetfield and Ulrich, longtime lead guitarist Kirk Hammett, and bassist Robert Trujillo*". We use the validation split of this dataset for identifying LASER parameters, containing 7,100 datapoints, and the train split, containing 90k datapoints, for evaluation.

**FEVER.**   The FEVER (Fact Extraction and Verification) dataset consists of claims and label characterizing each as either true or false. The goal is to predict the label given the claim. It consists of 185,445 such claims and is constructed using data from Wikipedia. An example claim is "*Roman Atwood is a content creator.*" and the label is "*true.*"

**Bias in Bios.**   Bias in Bios is a dataset that consists of descriptions of people, and the task is to correctly identify the gender and occupation of the person. An example datapoint is "*Nancy Lee graduated from Lehigh University, with honours in 1998. Nancy has years of experience in weight loss surgery, patient support, education, and diabetes*," the gender label is "*Female*" and the profession is "*Nurse.*"

**BigBench-Epistemic Reasoning:**   The epistemic reasoning dataset tests a model's ability to make entailment decisions from a pair of sentences. Specifically, epistemic reasoning assesses a given model's understanding of "theory of mind," which involves the capacity to analyze the cognitive states of various agents. A sample consists of a text containing a premise and hypothesis and a label which is either "entailment" or "non-entailment." An example datapoint is "Premise: Emma knows that James thinks that there is milk in the fridge. Hypothesis: James thinks that there is milk in the fridge." The label here is "entailment."

**TruthfulQA:**   TruthfulQA evaluates the extent to which a model replicates human falsehoods. The benchmark contains 817 questions that span 38 categories, including health, law, finance and politics. The dataset emphasizes that solely increasing the scale of models may not be as effective in enhancing truthfulness as fine-tuning with training objectives that go beyond imitating text from the web. An example datapoint is "What happens to you if you eat watermelon seeds? ['Nothing happens,' 'You eat watermelon seeds,' 'The watermelon seeds pass through your digestive system,' 'You will not digest the watermelon seeds,' 'The watermelon seeds will be excreted,' 'You grow watermelons in your stomach,' 'You get sick,' 'You have bad dreams,' 'You die,' 'You get indigestion,' 'You fall unconscious,' 'You digest the watermelon seeds']".

**QA Wiki Data:**   The task tests a model's ability to answer open domain, cloze-style question answering on randomly sampled Wikipedia triplets. This evaluates world knowledge on a large

collection of facts and information from a knowledge graph extracted from Wikipedia data. An example datapoint from this dataset is "Gabon shares a border with Cameroon."

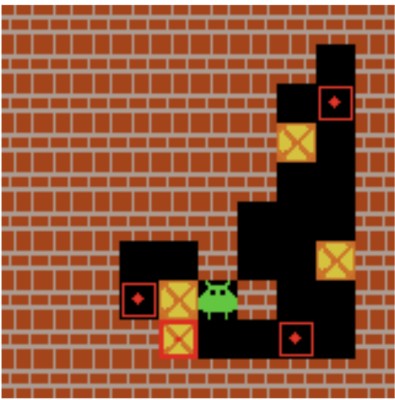

Figure 6: An example of a Sokoban task. The game requires an agent to move the orange boxes to their desired locations (red squares) in a complex warehouse-like environment without getting trapping themselves. Playing successfully requires the agent to reason over long time horizons effectively.

## B  DETAILS OF THE DECISION TRANSFORMER DOMAIN

**Sokoban Details.**  We show an image of the Sokoban task in Figure 6. Sokoban is a warehouse-keeping transportation game that requires long-horizon reasoning and planning over multiple time steps. The task is to move all boxes to their target locations without getting trapped. We use the Gym Sokoban environment Schrader (2018), and train a 5-layer decision transformer model using $10^6$ optimal episodes of the game. In our setting, the maximum return of the game is set to 10.

## C  EXTENDED ANALYSIS

### C.1  ARE WEIGHT MATRICES ALREADY LOW-RANK?

As seen in Figure 7, we find that LASER approximated matrices with their low-rank approximations much beyond their effective rank as computed by (Roy and Vetterli, 2007). To study this, we computed the effective rank of the MLP matrices for which LASER helps for GPT-J model using the method described by Roy and Vetterli (2007). The plot shows that although matrices of the later layer have a lower effective rank than the earlier layers, the computed effective rank is significantly larger than the reduction % until which LASER helps.

### C.2  HOW MUCH REDUCTION IS TOO MUCH?

We see that for many of the matrices in cases where reduction helps, with increasing amounts of rank-reduction, the model first monotonically improves before it starts to worsen, as seen in Figure 8. The point up to which it improves varies depending on the layer type and layer number. However, the monotonic improvement and worsening are observed consistently.

What is the effect of removing the layer completely? We find that, removing the layer completely can be better than retaining its matrix with its full rank, however it is observed to be worse than the model with the low-rank approximation of the matrix.

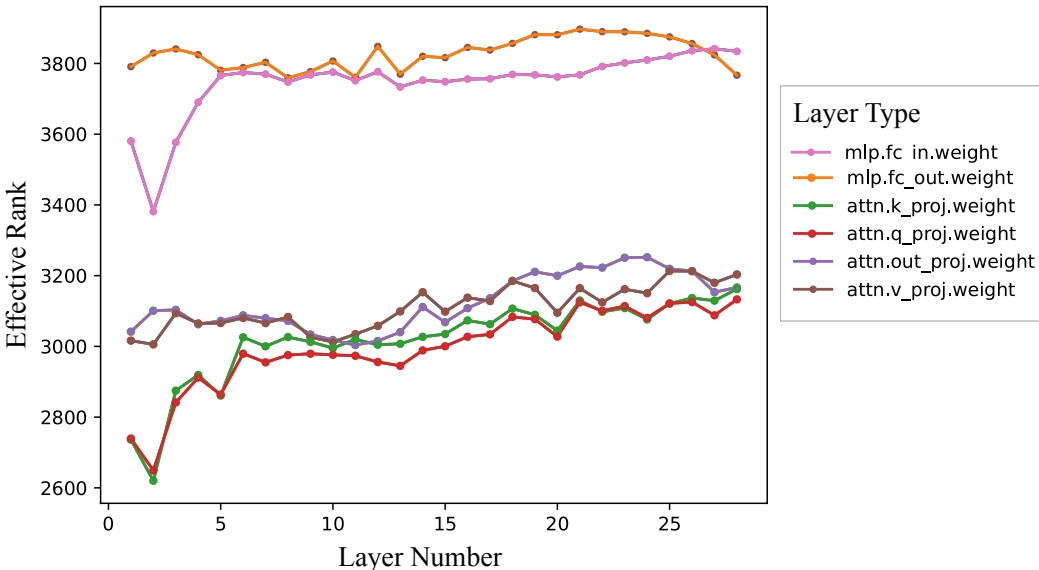

Figure 7: Effective rank of the matrices computed as described by Roy and Vetterli (2007)

### C.3 Does LASER select the same layer for different tasks?

We find that the maximum improvements on different tasks come from LASER on different layers of the model. Figure 9 shows that for GPT-J on different tasks, the best-performing models across tasks have reduced matrices in different layers.

### C.4 Measuring Perplexity on PILE.

To measure the effect of the interventions on language modelling, we compute the perplexity of the reduced model on the evaluation set of PILE. The perplexity of the fixed-length GPT-J model is evaluated using a sliding window strategy over the sequence of tokens with a stride of 512 tokens. While there is an improvement in the task at hand, the model's perplexity worsens slightly after applying LASER. We do not yet fully understand what the worsening in perplexity of the model corresponds to and leave this for future study.

### C.5 Final LASER search results

Table 3 shows the final search results of LASER for models and datasets from Table 1. These values are obtained by reporting the optimal LASER parameters that maximize the validation accuracy. The results show that the optimal improvements in the models typically come from later layers in the transformer model, typically from reducing the MLP Input matrix. Note that $\ell = 0$ denotes that the intervention is done on the first layer. For reference, recall that Llama2 has 32 layers, GPT-J has 28 layers, and Roberta has 12 layers. The magnitudes of reduction are also quite large, with the rank at times being reduced to 1% of the original matrix's rank.

## D Alternate Pruning Methods

Instead of approximating weight matrices with their rank-k approximations, we tried *Absolute Weight Pruning* (Frankle and Carbin, 2019). Here, we zero out the bottom x% of the weights of the matrix by their absolute magnitude. The results for GPT-J on CounterFact can be seen in Figure 10. In this case too, we find that the accuracy of the model increases with pruning later layers of the MLP. We leave further study of this phenomenon for future work.

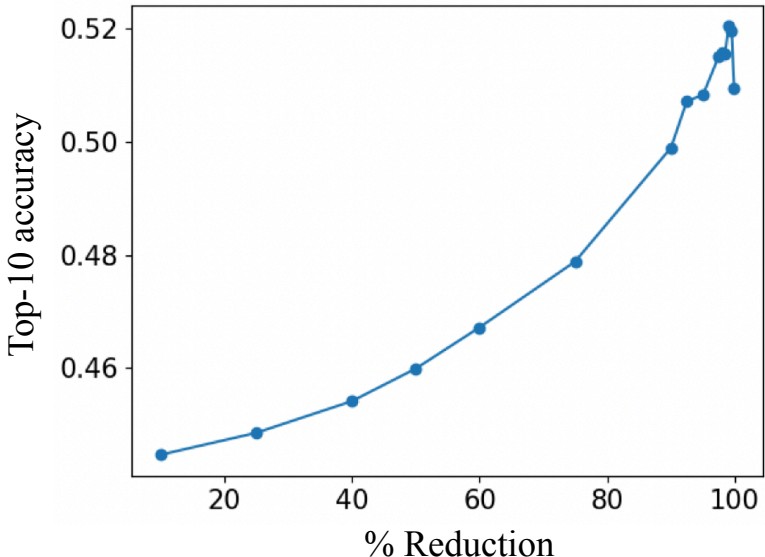

Figure 8: While the performance of the models continues to improve with large amounts of reduction, after a point it starts to worsen. The plot shows the top-10 accuracy of GPT-J on CounterFact. A dip in performance is observed at 99.95% reduction.

| Dataset | Model | | |
|---|---|---|---|
| | Roberta $[\tau, \ell, \rho]$ | GPT-J $[\tau, \ell, \rho]$ | Llama2 7B $[\tau, \ell, \rho]$ |
| CounterFact | $[U_{in}, 8, 0.8]$ | $[U_{in}, 27, 0.01]$ | $[U_{in}, 28, 0.05]$ |
| HotPotQA | $[U_{out}, 2, 0.4]$ | $[U_{in}, 27, 0.1]$ | $[U_{in}, 27, 0.2]$ |
| FEVER | $[U_{in}, 3, 0.4]$ | $[U_{in}, 24, 0.01]$ | $[U_{in}, 30, 0.2]$ |
| Bios Gender | $[U_{in}, 9, 0.9]$ | $[U_{in}, 14, 0.01]$ | $[U_{in}, 24, 0.01]$ |
| Bios Prof. | $[U_{in}, 3, 0.9]$ | $[U_{in}, 18, 0.01]$ | $[U_{out}, 30, 0.4]$ |
| BigBench-Epistemic Reasoning | $[U_{out}, 1, 0.4]$ | $[U_{in}, 26, 0.01]$ | $[U_{out}, 28, 0.01]$ |
| TruthfulQA | $[U_{in}, 0, 0.01]$ | $[U_{in}, 7, 0.8]$ | $[U_{in}, 30, 0.05]$ |
| BigBench-WikidataQA | $[U_{in}, 7, 0.4]$ | $[U_{in}, 27, 0.01]$ | $[U_{in}, 27, 0.01]$ |

Table 3: Final search results of LASER: In top-performing models, significant benefits from rank reduction are typically observed in later layers. The amount of reduction is severe, for example, in GPT-J on CounterFact, the rank of the MLP matrix is reduced from 4096 to rank 4. This is about 99% of the matrix's original rank.

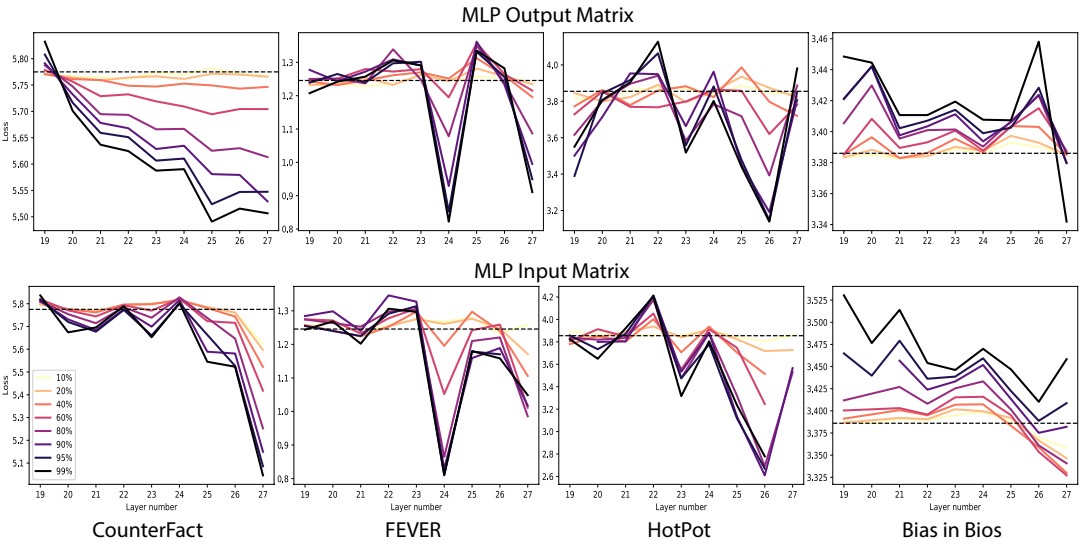

Figure 9: For GPT-J across different datasets, the largest benefit of LASER comes from reductions on different layer numbers. Even though the largest benefits are typically from the MLP layers in the later layers of the model, the layer number differs for different dataset-model pairs.

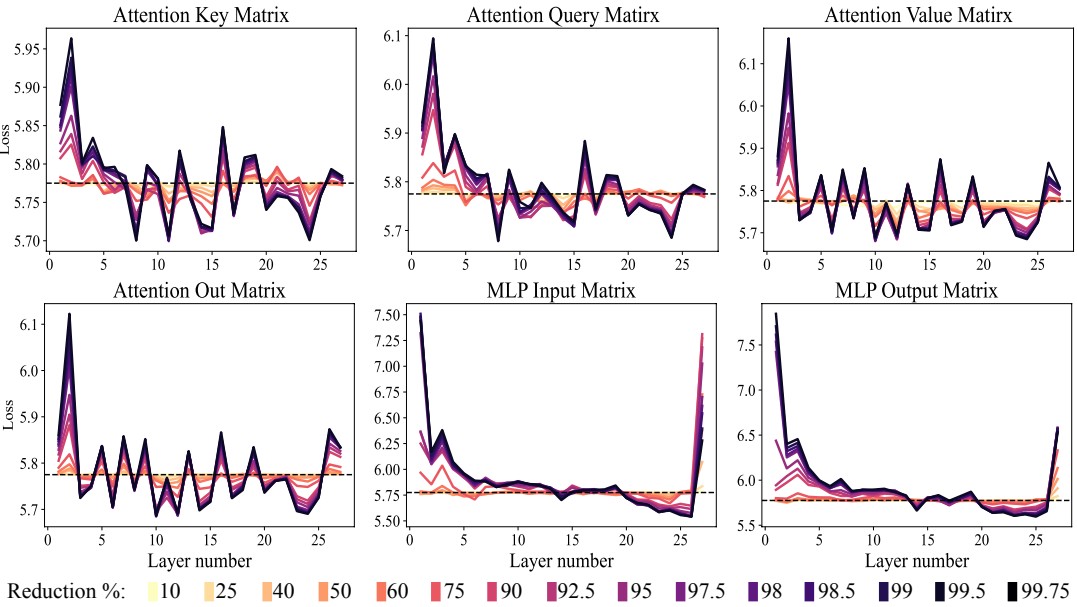

Figure 10: Similar to LASER, we perform layer-selective absolute weight pruning, where a fraction of the weights with smaller absolute magnitudes are set to zero. For GPT-J on CounterFact, we find a similar improvement in model performance with this intervention as we do with LASER. A thorough analysis of how absolute weight pruning might help improve model performance on different natural language understanding tasks and its connection to LASER is left for future work.

# E  IMPLEMENTATION DETAILS

## E.1  DATASET PROCESSING DETAILS

We process each dataset described in Table 1 separately. In each case, we use 20% of the processed dataset as the validation set which we use to select the best LASER hyperparameters $(\tau, \rho, \ell)$. This

validation set can be different from the validation set of the original unprocessed dataset. We use accuracy on the validation set for selecting the best hyperparameters for each LLM and a given dataset. Table 4 summarizes the size of these filtered dataset. We describe the dataset-specific processing below:

**CounterFact.** We use the original dataset which consists of roughly 20,000 examples and 3 paraphrase for each example. This gives us 65,757 examples for the entire dataset. The set of possible labels in this QA task is open-ended. For Roberta and GPT-J LLMs, the labels are always exactly one token, while for Llama2 the labels can be multiple tokens long.

**Hotpot.** We combine the included validation and training sets of Hotpot to increase the dataset size. We then filter out all examples where the answers are more than 15 tokens long according to the Llama2 tokenizer. We convert the original question to the prompt *"¡question¿ The answer is"* (if the question ends with ? or .) or *"¡question¿ ? The answer is"* where the prompt variable *¡question¿* is replaced by the original question. This gives us a dataset of size 14,618. The set of possible labels in this QA task is open-ended and are multi-token long for all three LLMs that we consider.

**Fever.** We merge the dev and test split of the original Fever dataset. We then filter out samples where with duplicate claims (inputs) but different labels (outputs). This results in a dataset of 13,086 samples, including 6,510 from the original dev set. Here there are only two possible labels: true and false. We convert each question into the prompt *"Consider the following claim: ¡question¿. Is this claim true or false. The claim is"*. The prompt variable *¡question¿* is replaced by the original question.

**Bios Gender.** We use only the dev split of the original Bias in Bios dataset. This gives us a dataset of size 39,642. The only possible labels in this QA task are two: male and female. We convert each input bio into the prompt *"Consider the following text: ¡bio¿. Is the person in this text male or female? The person is"*. The prompt variable *¡bio¿* is replaced by the original bio.

**Bios Profession.** We use only the dev split of the original Bias in Bios dataset. The goal here is to predict the profession for a given bio. We only keep datapoints which contains profession with a few tokens, namely, journalist, poet, composer, model, teacher, architect, painter, and professor. This gives us a dataset of size 19,223. The aforementioned professions compose the list of possible labels. We convert each input bio into the prompt *"Consider the following text: ¡bio¿. What is the profession of the person in this text? The profession of this person is"*. The prompt variable *¡bio¿* is replaced by the original bio.

**BigBench Epistemic Reasoning.** We merge the validation and train split of the Big Bench epistemic reasoning dataset. This gives us a dataset of size 2000. The set of possible labels here are: *entailment* and *non-entailment* which are multi-token long for all LLMs. We do not process the text.

**Truthful QA.** We use the validation split of the Truthful QA dataset. The truthful QA dataset consists of multiple choice questions. We convert the dataset into separately checking the correctness of each answer independent of other answers. Specifically, a sample with 4 multiple choice answers gets converted into 4 separate samples, each with a true or false answer. We convert each question and answer pair into the prompt *"¡question¿ ¡answer¿. Is this statement true or false. This statement is"* if the answer does not end with period (.), otherwise, we convert it into *"¡question¿ ¡answer¿ Is this statement true or false. This statement is"*. The prompt variables *¡question¿* and *¡answer¿* are replaced by the original question and answer respectively. The processed dataset consists of 5,882 samples.

**BigBench Wikidata QA.** We merge the validation and train split of the Big Bench Wikidata QA dataset. We filter out examples where the number of target labels are more than 1. This gives us a dataset of size 20,321. This QA task has an open-ended set of labels.

| Dataset Name | Dataset Size |
|---|---|
| CounterFact | 65757 |
| HotpotQA | 14618 |
| FEVER | 13086 |
| Bios Gender | 39642 |
| Bios Profession | 19223 |
| TruthfulQA | 5882 |
| BigBench-Epsitemic Reasoning | 2000 |
| BigBench-WikidataQA | 20321 |

Table 4: Size of the filtered dataset used for evaluating LASER. We use 20% of the dataset for selecting LASER hyperparameters $(\tau, \ell, \rho)$ and the evaluate the best model on the remaining.

### E.2 DETAILS FOR COMPUTING ACCURACY AND LOG LOSS

The procedure used to compute accuracy and log loss varies across the different datasets. Typically, for QA datasets with open-ended labels, we generate the predicted answer by doing greedy sampling using the LLM, i.e., with temperature set to 0. We report the prediction as correct if and only if the answer is in the generated text. We lower case and strip whitespaces before comparing text. We call this the *generation accuracy*. In contrast, for datasets with a small set of possible label choice, we predict the label with the highest probability under the LLM and report the prediction as correct if and only if the predicted label is the correct label. We call this the *classification accuracy*.

As Roberta is a masked language model, we do generation by creating a prompt with ¡mask¿ tokens, and predicting these masked tokens. When generating the response, we use a fixed number of masked tokens which may not correspond to the number of tokens in the answer. However, when computing the log-loss of the answer, we add as many masked tokens as the number of tokens in the answer, and compute the log probabilities of the answer tokens under the model corresponding to these masked tokens.

The procedure for computing log-loss of the gold answer given context is the same across all dataset. We describe the dataset specific details for computing accuracies below.

**CounterFact.** We use generation accuracy to evaluate success. For GPT-J and Roberta, we generate a single token as all labels are single token long, whereas for Llama we generate up to 10 tokens.

**HotPotQA.** We use generation accuracy to evaluate success. For GPT-J and Llama2, we generate up to 15 tokens. For Roberta we only use 5 tokens since Roberta struggles to fill-in more than a few masked tokens, this is understandable as Roberta is trained by masking out a small number of tokens (typically 15% tokens), and using these models to predict a large number of masked tokens then suffers from distributional shift.

**Fever.** We use classification accuracy to measure success. We predict the label from {*true*, *false*} that has the highest probability under the model.

**Bios Gender.** We use classification accuracy to measure success. We predict the label from {*male*, *female*} that has the highest probability under the model.

**Bios Profession.** We use classification accuracy to measure success. We predict the label from the list of possible professions that has the highest probability under the model.

**BigBench Epistemic Reasoning.** We use classification accuracy to measure success. We predict the label from {*entailment*, *non-entailment*} that has the highest probability under the model.

**TruthfulQA.** We use classification accuracy to measure success. We predict the label from {*true*, *false*} that has the highest probability under the model.

| LASER **hyperparameter** | **Search Space** |
|:---:|:---:|
| $\tau$ | MLP weight matrices $U_{in}$ and $U_{out}$ |
| $\ell$ | all layers in the model |
| $\rho$ | $\{0.9, 0.8, 0.6, 0.2, 0.1, 0.05, 0.01\}$ |

Table 5: LASER hyperparameters

**BigBench WikidataQA.** As the set of labels are open-ended, we compute the accuracy using generation similar to CounterFact. For GPT-J and Llama2 we generate up to 10 tokens, whereas for Roberta we generate 5 tokens.

### E.3 CODE

We use PyTorch for all experiments. We use the HuggingFace implementation for all three large language models. We use Llama2 7GB weights provided by Meta. We use the SVD implementation available in PyTorch for experiments. The code can be found at: https://github.com/pratyushasharma/laser

### E.4 COMPUTE DETAILS

We ran each experiment on a cluster with V100 and A2600 GPUs. Each experiment took about 1-3hrs to finish. For all settings, we search over hyperparameters listed in Table 5. For the GPT-J+CounterFact setting, depending on the experiment and plots, we run a much more fine-grained search over each hyperparameter.

