# OpenReview forum: "The Truth is in There: Improving Reasoning in Language Models with Layer-Selective Rank Reduction"
_ICLR.cc/2024/Conference — ICLR 2024 poster_

### Official Review · Reviewer_Zz8q · 2023-10-28

**Soundness:** 4 excellent
**Presentation:** 4 excellent
**Contribution:** 3 good
**Rating:** 8
**Confidence:** 3

**Summary:**

This paper introduces a layer-selective rank-reduction method called LASER. The authors demonstrate that the performance in open-ended question answering is improved when rank-reduction is applied to the specific weight matrix. Moreover, they confirm consistent performance enhancements in tasks on non-text domains such as policy learning and image classification. Additionally, through analysis, it has been observed that high-order components contain factually incorrect knowledge which degrades question answering performance.

**Strengths:**

- The authors conduct extensive experiments with respect to layer number, parameter type, and rate reduction to identify setups that lead to performance improvement.
- The authors provide interesting observations such as a correlation between rank reduction and question answering accuracy.
- The authors demonstrate that the proposed method can also be applied to various domains such as image classification.

**Weaknesses:**

- Analysis on other text domain tasks such as reading comprehension could provide further insights.

**Questions:**

- In Figure 3(c), what is the number of originally correct and answer-corrected datapoints?
- It appears that there is a significant improvement in overall performance with GPT-J compared to LLama2 or Roberta. What could be the underlying cause of the result?

---

> ### Author Response · Authors · 2023-11-21
> **Clarifications and results on 3 datasets, including the task of reading comprehension.**
>
> We thank you for your feedback.
>
> >Analysis on other text domain tasks such as reading comprehension could provide further insights.
>
> **Results on New Datasets:** We have added analysis on Epistemic Reasoning (logic and reading comprehension), TruthfulQA (factuality), and QA Wiki Data (world knowledge). The Epistemic Reasoning dataset is provided by Big Bench and was developed specifically to measure reading comprehension and reasoning. Please see Table 4 in the Appendix and the general response. All edits are written in red.
>
> As with other datasets, we find that LASER offers notable boosts in performance across model architectures. We believe that **improvements across 8 datasets and multiple LLMs and tasks provide strong evidence for using LASER.**
>
> >In Figure 3(c), what is the number of originally correct and answer-corrected data points?
>
> **Clarification:** We apologize for not defining these terms clearly. “Originally correct” describes samples that are correctly classified even without any intervention.  “Answer-corrected” refers to questions that the model gets correct only after intervening with LASER (i.e., they were not correctly classified before executing LASER). We have edited the text to clarify this in the new version of the paper.
>
> >It appears that there is a significant improvement in overall performance with GPT-J compared to LLama2 or Roberta. What could be the underlying cause of the result?
>
> **Possible Causes:** Thank you for the question and observation. Although the improvements in GPT-J are more pronounced than in LLAMA2 or Roberta,  there are some domains where improvements in LLAMA2 (FEVER) and Roberta (Bias in bios) are more significant than in GPT-J as well. We consider a thorough analysis of this to be outside the scope of this work, but there are several possible causes that we believe merit investigation, including the capacity of the model, the amount of training data, and the particulars of the optimization procedure. We have added this as an important direction for future research in the current draft.

---

> > ### Comment · Reviewer_Zz8q · 2023-11-22
> >
> > I have read the comment including additional results, and raised my score.

---

### Official Review · Reviewer_4cdi · 2023-10-30

**Soundness:** 2 fair
**Presentation:** 3 good
**Contribution:** 2 fair
**Rating:** 3
**Confidence:** 4

**Summary:**

This work discussed a traditional idea, i.e. the low-rank approximation using SVD, for language model compression. Its major observation is that using a low-rank approximation on the MLP layer of transformers can even improve the downstream performance.
This observation is verified across different tasks and different transformer models.

**Strengths:**

-The use of low-rank approximation should be an effective and general way to allow the models to obtain more robust generalization abilities, while being more computationally efficient in the inference.
-The authors tried very hard to demonstrate the major observation by showing the results across different transformer models, which are actually not even for language modeling tasks.

**Weaknesses:**

Albeit the strengths above, I would like to say the major weakness of this work is that it draws its conclusion not very rigorously:
-Current LLMs are often evaluated from multiple aspects including their reasoning abilities such as commonsense reasoning, world knowledge, reading comprehension etc, as well as their language generation abilities such as multilingual ability etc. And each aspect contains well-known benchmark datasets for the evaluation, such as MNLU, AGIEval and BBH. However, this work uses none of them. Therefore, I am not convinced that this robust performance can be achieved across all the above-mentioned aspects.
-The authors do not provide the final search results of rank reduction, i.e. the layers selected for compressing and the reduced rank, in the final performance in Table1, 2 & 3. It is very important to provide these results to show that the selected model is indeed in a reduced rank.

**Questions:**

I find that the dimension of GPTJ is 4096, which should be $d$ in your notation. So in Fig2, what is the rank of Reduction 99.5%/99.75% and others with .5%? (4096*0.0025=10.24, not an integer?)

The used CounterFact is very similar to the table-to-text generation task (but for a qa task), which is not a frequently used dataset to test even the QA and factual/world-knowledge reasoning performance of LLM. Any reason for choosing the dataset?

---

> ### Author Response · Authors · 2023-11-21
> **Additional Results and Important Clarifications.**
>
> Thank you for your review.
>
> >This work discussed a traditional idea, i.e., the low-rank approximation using SVD, for language model compression.
>
> **Clarification:** We want to emphasize that the focus of the paper is not to compress models but to demonstrate instead that selective rank-reduction interventions can offer a significant boost in terms of predictive performance.
>
> We also emphasize that most past works (Lv et al., 2023; Hajimolahoseini et al., 2021; Yu et al., 2017) that do a low-rank approximation of neural networks or transformers perform low-rank approximation on every weight matrix and/or every layer of the model. In contrast, LASER is layer-selective, meaning we intervene in selective weight types (e.g., MLP input matrix ($U_{in}$)) and selective layer numbers. In fact, doing low-rank approximation unilaterally across layer numbers and layer types often leads to a significant reduction in performance. We believe this subtle difference with past work is an important finding.
>
> To our knowledge, papers that use a low-rank approximation of matrices for model compression at best only obtain roughly accuracy-preserving models. We instead demonstrate that selective reduction of this sort can offer significant performance improvements.
>
> >Current LLMs are often evaluated from multiple aspects including their reasoning abilities such as commonsense reasoning, world knowledge, reading comprehension, etc...such as MNLU, AGIEval, and BBH...this work uses none of them...
>
> **Additional Experiments on 3 new datasets:** We evaluate LASER on 5 benchmarks and 3 LLMs, which gives 15 different setups in total. The sizable improvements given by LASER across these setups demonstrate a clear and robust result.
> However, we agree that evaluating LASER on more setups will bolster our arguments. In light of this, we have added results on additional new datasets, including Epistemic Reasoning (from Big Bench Hard) and QA Wikidata (from Big Bench) alongside the Truthful QA benchmark. Of these, the big bench hard was specifically suggested in the review. We were unable to locate the MNLU dataset.
>
> These new results are in the general response and in Table 4, Appendix C of the paper. All relevant edits are written in red. These results show similar notable improvements as in our main paper. For example, we notice a 14% point increase in test accuracy on QA Wikidata with GPTJ and an 18.6% point increase in test accuracy on Epistemic reasoning with Llama2. We also notice that, similar to Table 1 in our main paper, some setups have more modest gains of 0.5-2%. Understanding which datasets have more improvements and why is an interesting avenue for future work.
>
> We will release our code and provide results on more datasets in the future, but we believe that **positive results across 8 datasets and multiple LLMs and tasks establish the general usefulness of LASER and that these results cannot be a coincidence.**
>
> >The authors do not provide the final search results of rank reduction...It is very important to provide these results to show that the selected model is indeed in a reduced rank
>
> **Final Search Results:** Thank you for pointing this out. We have added the final search results of the reduced ranks in Table 5 and Table 6 of the updated paper.  As noted in the paper, the largest gains typically occur when intervening in higher layers of the LLM. For reference, Llama2 has 32 layers, GPTJ has 28 layers, and Roberta has 12 layers.
>
> >I find that the dimension of GPTJ is 4096, which should be in your notation. So in Fig2, what is the rank of Reduction 99.5%/99.75% and others with .5%? (4096*0.0025=10.24, not an integer?)
>
> **Rounding Clarification:** As mentioned in Section 4, last line of paragraph 1: “We replace it with a rank ⌊ρ · d⌋-approximation.”  Here, “⌋” represents the floor of the number, i.e., the number is rounded down to the nearest integer value.
>
> >The used CounterFact is very similar to the table-to-text generation task (but for a QA task), which is not a frequently used dataset to test even the QA and factual/world-knowledge reasoning performance of LLM. Any reason for choosing the dataset?
>
> **Justification for using Counterfact:** The CounterFact dataset (introduced by Meng et al. 2022 ) is derived from the PARAREL dataset (Elzar 2021) and WikiData. This dataset is well-cited and commonly used for research on LLM understanding, and this was the main reason why we picked it. Including this dataset afforded the measurement of robustness to paraphrases with and without LASER (See Section 5.1 para 3). This study deepened our understanding of LASER’s robustness to paraphrases. Additionally, we provide results on 7 other datasets, including 3 new datasets that we added in this rebuttal. Specifically, we have also included the effect of LASER on WikiData QA, a similar benchmark from Big Bench.

---

> > ### Comment · Reviewer_4cdi · 2023-11-22
> >
> > Again, I appreciate the authors trying very hard to demonstrate their observations by adding more results to their responses. However, as Reviewer F88a says, this paper presents a surprising result, I still want to hold back my score. The reason is that the authors try to say that this low-rank approximation and the resulting improvement could generalize well to a wide range of transformation models. However, the backbone models here are very limited. GPTJ and Roberta may not be the most suitable base models to validate currently. Also, if we want to demonstrate even a very simple modification to work well on current LLMs (say Llama alone), we need to evaluate extensive benchmark dataset collections, as I mentioned (MMLU, BBH, AGIEval and many others), and each of them includes multiple datasets (say BBH has 27 datasets). So, there is still a long way to go before reaching the conclusion that this work draws here. And I will be very pleased if someday the authors can convince the community to incorporate this idea as a standard module for Transformer.

---

### Official Review · Reviewer_F88a · 2023-10-30

**Soundness:** 4 excellent
**Presentation:** 4 excellent
**Contribution:** 4 excellent
**Rating:** 10
**Confidence:** 4

**Summary:**

LLMs are usually considered “the larger the better”, but this paper presents a surprising result: it is often possible to improve the performance of LLMs by simply removing higher-order components of their constituent weight matrices in the MLP layers. This paper presents this rank reduction method, LASER, that removes the components in the {Q,K,V,O} matrices that have smaller singular values (i.e., those higher-order components).

This paper finds that the effects of reduction is not uniform across layers. The performance degradation can be found by reducing early layers, while significant performance benefits are available, often by pruning the later layers. This effect is the most obvious in the MLP output, and is also observable in the k, q, v matrices.

This paper further dives into studying what types of facts are recovered by rank reduction, and finds that the facts recovered on rank reduction are most likely those infrequently present in the data.

Why are the higher-ordered components noisy? And what are the higher-ordered components computing? This paper approximates the final weight matrix using its higher-ordered components, and analyze how the model’s behavior changes on the datapoints that GPT-J’s lower-order components lead to incorrect outputs. They find that the model predicts incorrect entities of the same semantic type as the correct answer. As more lower-ordered components are included, the output changes to predicting common word tokens.

With additional experiments (on text domains including QA and non-text-domains including learning policies, images), this paper studies the generalizability of the findings.

**Strengths:**

- This paper is well-written and easy to read.
- The experiments are designed thoughtfully, and nicely supports the hypothesis.
- The findings are important for both the understanding and the developments of better models in the future.

**Weaknesses:**

I do not see obvious weaknesses in this paper. There is a typo: Table 3 needs a horizontal line at the bottom.

**Questions:**

Seems like the MLP layers are the key components in storing the noise vs storing the “useful inductive biases”. I wonder if some structural choices (e.g., different position embedding methods, different activation functions, number of heads, etc.) can affect Transformer’s low-rank vs high-rank component behavior as well.

---

> ### Author Response · Authors · 2023-11-21
> **Thank you. Added additional discussion and results.**
>
> We thank you for your encouraging remarks.
>
> We have added the missing horizontal line in Table 3 and thank you for pointing it out.
>
> _We have also added results on more datasets, which also show notable gains due to LASER and further support our main argument (see Table 4 in Appendix F and in the general response)._ We have also added experimental details in the Appendix. All edits are indicated by red text.
>
>
> >Seems like the MLP layers are the key components in storing the noise vs storing the “useful inductive biases”. I wonder if some structural choices (e.g., different position embedding methods, different activation functions, number of heads, etc.) can affect Transformer’s low-rank vs high-rank component behavior as well.
>
> Examining how structural choices in model design affect the low-rank and high-rank components of the MLP layers is an intriguing question that could advance our understanding of this phenomenon and of transformer models in general. While we consider a thorough investigation of this question to be outside the scope of this work, and it will require training models with different structural choices from scratch, we have included a discussion of this in our paper to contextualize our results in light of this discussion (please see Appendix F).

---

> > ### Comment · Reviewer_F88a · 2023-11-22
> > **Reviewer reply**
> >
> > Thanks for the response. I'm happy to keep my original score.

---

### Author Response · Authors · 2023-11-21
**General Response: Revised paper uploaded. Results on 3 new datasets and additional discussion.**

We thank the reviewer for their feedback.

We are glad the reviewers found that the experiments are designed thoughtfully [F88a] and are extensive [Zz8q]. Additionally, they noted that the method provides an effective and general way to allow the models to obtain more robust generalization [4cdi] and that the findings are important for both the understanding and the development of better models in the future [F88a].

We have revised the paper to include additional experimental details and results on 3 new datasets. This includes Epistemic Reasoning from Big Bench Hard (BBH) (logic and reading comprehension), TruthfulQA (language model truthfulness), and QA Wiki Data from Big Bench (world knowledge), all showing sizable improvements with LASER. In particular, **Big-bench hard datasets were recommended by Reviewer 4cdi, and reviewer Zz8q suggested adding results on additional text tasks such as reading comprehension.**

The new results are in Table 4 in the Appendix of the updated paper and are also pasted below for convenience. **Each demonstrates notable improvements with LASER, similar to the results in Table 1 of the original paper.**

*Table 4: Effect of LASER on additional datasets*
| Model | Epistemic Reasoning | TruthfulQA | QA Wiki Data|
| -------- |-------- |-------- | -------- |
|  GPT-J     |   37.1%  | 54.9% | 51.8%
| **GPT-J with LASER**    |   **38.3%**  | **55.6%** |**65.9%**
| LLAMA2   |    44.8%  | 50.5%|59.5%
| **LLAMA2 with LASER**   |    **63.4%**   |**56.2%**|**62%**

We've additionally corrected typos and added more experimental details as per the reviewers' suggestions.

---

### Meta-Review · Area_Chair_LoHQ · 2023-12-07

**Metareview:**

This paper proposes a method called LASER that selectively replaces the weight matrices in Transformers’ MLP layers with their low-rank approximations. The paper shows that (surprisingly) such pruning can actually improve model performance, as demonstrated on a range of tasks including open-ended QA, policy learning, and image classification.

The reviewers generally found that such unexpected results (pruning leads to improved performance) interesting, and appreciate the thorough experiments. However, reviewer 4cdi is less convinced by the conclusions due to lack of results on more mainstream benchmarks (such as MMLU and BBH). Reviewers would also hope to see more analyses to give intuitions why the proposed method help.

**Justification For Why Not Higher Score:**

More analysis is needed for convincingly provide insights for why the method helps.

**Justification For Why Not Lower Score:**

The proposed method is simple and interesting. The paper shows surprising findings (pruning leads to improved performance), which can be useful for the research community.

---

### Decision · Program_Chairs · 2024-01-16

Accept (poster)